# Mapping the Serum Proteome of COVID-19 Patients; Guidance for Severity Assessment

**DOI:** 10.3390/biomedicines10071690

**Published:** 2022-07-13

**Authors:** Estefanía Nuñez, Irene Orera, Lorena Carmona-Rodríguez, José Ramón Paño, Jesús Vázquez, Fernando J. Corrales

**Affiliations:** 1CIBER de Enfermedades Cardiovasculares (CIBERCV), 28029 Madrid, Spain; estefania.nunez@cnic.es; 2Cardiovascular Proteomics Laboratory, Centro Nacional de Enfermedades Cardiovasculares (CNIC), 28029 Madrid, Spain; 3Proteomics Research Core Facility, Instituto Aragonés de Ciencias de la Salud (IACS), 50009 Zaragoza, Spain; iorera.iacs@aragon.es; 4Functional Proteomics Laboratory, Centro Nacional de Biotecnología (CSIC), 28049 Madrid, Spain; lcarmona@cnb.csic.es; 5Division of Infectious Diseases, Hospital Clínico Universitario, IIS Aragón, Ciberinfec, 50009 Zaragoza, Spain; jrpanno@salud.aragon.es

**Keywords:** SARS-CoV-2, COVID-19, proteomics, severity prognostics, biomarkers

## Abstract

Coronavirus disease 2019 (COVID-19) is caused by severe acute respiratory syndrome coronavirus 2 (SARS-CoV-2), whose outbreak in 2019 led to an ongoing pandemic with devastating consequences for the global economy and human health. According to the World Health Organization, COVID-19 has affected more than 481 million people worldwide, with 6 million confirmed deaths. The joint efforts of the scientific community have undoubtedly increased the pace of production of COVID-19 vaccines, but there is still so much uncharted ground to cover regarding the mechanisms of SARS-CoV-2 infection, replication and host response. These issues can be approached by proteomics with unprecedented capacity paving the way for the development of more efficient strategies for patient care. In this study, we present a deep proteome analysis that has been performed on a cohort of 72 COVID-19 patients aiming to identify serum proteins assessing the dynamics of the disease at different age ranges. A panel of 53 proteins that participate in several functions such as acute-phase response and inflammation, blood coagulation, cell adhesion, complement cascade, endocytosis, immune response, oxidative stress and tissue injury, have been correlated with patient severity, suggesting a molecular basis for their clinical stratification. Eighteen protein candidates were further validated by targeted proteomics in an independent cohort of 84 patients including a group of individuals that had satisfactorily resolved SARS-CoV-2 infection. Remarkably, all protein alterations were normalized 100 days after leaving the hospital, which further supports the reliability of the selected proteins as hallmarks of COVID-19 progression and grading. The optimized protein panel may prove its value for optimal severity assessment as well as in the follow up of COVID-19 patients.

## 1. Introduction

Coronavirus disease of 2019 (COVID-19) is caused by the severe acute respiratory syndrome coronavirus 2 (SARS-CoV-2) that has infected more than 551 million people and accounts for more than 6 million deaths worldwide since late 2019 (https://covid19.who.int/, 18 October 2021). Since COVID-19 was declared a pandemic by WHO, the scientific community has mobilized significant resources to unravel the mechanisms of SARS-CoV-2 infection and replication [1,2].

SARS-CoV-2 belongs to the enveloped positive-sense RNA B Betacoronavirus family [3] that uses angiotensin-converting enzyme-2 (ACE-2) as its primary entry receptor [4] upon interaction with the viral spike protein (S protein). After entry, the polycistronic viral RNA is immediately translated by the host ribosome and polyproteins Orf1a and Orf1b are produced. These polyproteins are processed by viral proteases to produce 16 nonstructural proteins (Nsps). Then the full-length RNA is replicated through a negative sense intermediate and accessory proteins are synthetized [5]. Virion assembly occurs in the ER–Golgi intermediate compartment and mature virions are released via the lysosomal exocytosis pathway [6].

The clinical presentation of COVID-19 is highly variable ranging from asymptomatic infection to severe clinical conditions that may include acute respiratory distress and exacerbated inflammation leading to respiratory failure, multi-organ failure and ultimately death [7,8,9]. This complex scenario contributes to the difficulty of containing SARS-CoV-2 spread and justifies the tremendous efforts that have been, and are currently, devoted to developing diagnostic methods, vaccines (https://www.who.int/publications/m/item/draft-landscape-of-covid-19-candidate-vaccines, accessed on 18 October 2021) and treatments (https://www.who.int/publications/i/item/WHO-2019-nCoV-therapeutics-2021.3, accessed on 18 October 2021). In this regard, the understanding of the interaction between SARS-CoV-2 and the host cell is of utmost relevance.

Large-scale “omics” approaches are powerful technologies that have provided detailed genetic and genome structure information about SARS-CoV-2, including its homologies with other coronaviruses and variants. Once this information is available, the next step towards the elucidation of the infection mechanisms of this novel pathogen, is to analyze the viral and cell proteomes as well as their interactions. Proteins are the effectors of most cellular functions and the main components targeted by drug development programs. Proteomics can be defined as the set of analytical resources (liquid chromatography in combination with mass spectrometry and more recently affinity reagent-based methods) allowing the study of proteins and proteomes in their whole complexity. In the context of the current pandemic, this emerging discipline is being used to combat COVID-19 from many different perspectives. The identification of specific panels of circulating proteins, whose abundance is regulated along the progression of the disease, has provided valuable information to stratify COVID-19 patients according to their expected outcome [10,11,12,13] and point to drug repurposing strategies [14]. Structural proteomics is playing a central role in defining the structure of viral proteins and virions [15,16,17], as well as characterizing new generation therapeutics such as monoclonal antibodies [18,19]. Since proteins are the central layer of information transfer from the genome to the phenotype, deep proteomics studies will delineate the systems-wide perturbations occurring in infected organisms. This understanding of the pathogen-host interaction will pave the way to build up defenses against COVID-19 and future pandemics.

In this study, serum samples from 72 Spanish COVID-19 patients have been analyzed by shotgun liquid chromatography-electrospray-tandem mass spectrometry (nanoLC-ESI-MS/MS). Our results allowed the identification of a panel of proteins whose abundance changes in serum correlate with patient age and disease severity. These findings were verified in a second cohort containing 84 individuals by targeted MS/MS. Moreover, the regulated proteins returned to normal values upon resolution of the viral infection, further supporting the association of these proteins and the disease phenotype.

## 2. Materials and Methods

### 2.1. Human Samples

Human serum samples and data from COVID-19 patients were provided by the Biobank of the Aragon Health System (PT20/00112), integrated in the Spanish National Biobanks Network and they were processed following standard operating procedures with the appropriate approval of the Ethics and Scientific Committees. The present study was approved by the human research review committee of the Spanish Research Council, and was conducted in compliance with the ethical standards formulated in the Helsinki Declaration of 1996 (revised in 2000), upon obtention of the informed consent from all patients.

### 2.2. Sample Preparation

Serum aliquots (5 µL) of each individual (corresponding to about 300 µg of protein), were mixed in a 96-well multiplate with 5 µL of a buffer containing 50 mM Tris, 2% sodium dodecyl sulphate (SDS) and were heated at 56 °C for 30 min for virus inactivation. Thereafter, 100 mM dithiothreitol (DTT) was added, and samples were boiled at 100 °C for 5 min for protein denaturation. Then, proteins were subjected to filter-aided digestion using 96-well plates (AcroPrep™ 96-well Filter Plates, PALL, Port Washington, NY, USA) coupled to a vacuum manifold (PALL) according to manufacturer’s instructions. Briefly, 200 µL of urea were added to each sample to dilute SDS and cysteine residues were blocked with 50 mM iodoacetamide for 1h at 37 °C in darkness. After two washes with 100 µL urea followed by two washes with 100 µL of 100 mM ammonium bicarbonate pH 8.8 (AB), proteins were digested with trypsin (1:30-trypsin:protein, Promega, Madison, WI, USA) O/N at 37 °C. After protein digestion, peptides were eluted from the filter in two different steps using 40 µL of AB and 50 µL of 500 mM NaCl, respectively. Peptides were acidified with 25% trifluoroacetic acid (TFA) to a final concentration of 1%, desalted with Oasis cartridges (Waters, Milford, MA, USA) following manufacturer’s instructions, dried with a Speed-Vac and stored at −20 °C.

### 2.3. Multiplexed Isobaric TMT Labeling

Peptides were dissolved in 70 µL of 100 mM Triethylammonium bicarbonate (TEAB) buffer, and peptide concentration was measured using DirectDetect Infrared Spectrometer (Merck, Darmstadt, Germany). From each sample, 50 µg of peptides were subjected to multiplexed isobaric labeling with 10-plex tandem mass tags (TMT) reagents (Thermo Fisher Scientific, Waltham, MA, USA) following manufacturer’s instructions with minor changes. Each TMT experiment contained 9 individuals and one channel was reserved for reference internal standard samples created by pooling serum samples from all individuals of the cohort. Peptides were labeled with 0.8 mg of TMT reagents, previously reconstituted in 42 µL of acetonitrile, at room temperature for 1 h. Reaction was stopped with 0.5% TFA after 10 min incubation. After labeling, peptides from all the samples in each TMT experiment were mixed in the same tube, concentrated in a Speed-Vac, acidified with 25% TFA to a final concentration of 1% and desalted with Oasis cartridges (Waters, Milford, MA, USA). Aliquots of 1/10 (in volume) were reserved for direct MS analysis (without peptide fractionation) and the remaining peptides were saved for further peptide fractionation before MS analysis. Labeled peptide aliquots were dried-down and stored at −20 °C.

### 2.4. Peptide Fractionation

Peptides were fractionated using the high pH reversed-phase peptide fractionation kit (Thermo Fisher Scientific, Waltham, MA, USA) according to manufacturer’s instructions. Briefly, cartridges were washed with 50% and 100% acetonitrile (ACN), and equilibrated with 0.1% of TFA. A total of 100 µg of peptides were resuspended in TFA 0.1% and loaded into the cartridges. Peptides were then eluted into five fractions with increasing amounts of ACN: Fr1 (12.5% ACN), Fr2 (15% ACN), Fr3 (17.5% ACN), Fr4 (20% ACN) and Fr5 (50% ACN). Fractions were Speed-vac dried and stored at −20 °C until MS analysis.

### 2.5. LC-MS Analysis

Labeled peptide samples were analyzed using an Ultimate 3000 HPLC system (Thermo Fisher Scientific, Waltham, MA, USA) coupled via a nanoelectrospray ion source (Thermo Fisher Scientific, Bremen, Germany) to a Q Exactive HF mass spectrometer (Thermo Fisher Scientific, Waltham, MA, USA). C18-based reverse phase separation was performed using a PepMap 100 C18 5 μm 0.3 × 5 mm as trapping column (Thermo Fisher Scientific, Waltham, MA, USA) and a PepMap RSLC C18 EASY-Spray column 50 cm × 75 μm ID as analytical column (Thermo Fisher Scientific, Waltham, MA, USA). Peptides were loaded in buffer A (0.1% formic acid in water (*v*/*v*)) and eluted with an acetonitrile gradient consisting of 0–21% buffer B (100% ACN, 0.1% formic acid (*v*/*v*)) for 300 min and 21–90% B for 5 min at a flow rate of 200 nl/min. Mass spectra were acquired in a data-dependent manner. MS spectra were acquired in the Orbitrap analyzer using full ion-scan mode with a mass range of 400–1500 mass-to-charge (*m*/*z*) and 70,000 FT resolution. The automatic gain control target was set at 2 × 10^5^ with 50 ms maximum injection time. MS/MS was performed using the top-speed acquisition mode with 3s cycle time. HCD fragmentation was performed at 30% of normalized collision energy and MS/MS spectra were analyzed at a 60,000 resolution in the Orbitrap.

### 2.6. Protein Identification

For peptide identification MS/MS spectra were searched with the SEQUEST HT algorithm implemented in Proteome Discoverer 2.1 (Thermo Scientific) against a Uniprot database comprised human protein sequences (July 2014, 157430 entries), using trypsin digestion with a maximum of two missed cleavages, using Cys carbamidomethylation (57.021464 Da) and TMT labeling at N-terminal end and Lys (229.162932 Da) as fixed modifications, and Met oxidation (15.994915) as dynamic modification. Precursor mass tolerance was set at 800 ppm and fragment mass tolerance at 0.03 Da; precursor charge range was set to 2 to 4. Results were analyzed using the probability ratio method [20] and the false discovery rate (FDR) was calculated based on the search of results against the corresponding decoy database using the refined method [21], with an additional filter for precursor mass tolerance of 15 ppm [22]. An FDR of 1% was used as criterion for peptide identification.

### 2.7. Protein Quantification and Statistical Analysis

Protein quantification and statistical and systems biology analysis were performed using the models previously developed in our laboratory [23,24,25] with the SanXoT software package [26]. Quantitative information was extracted from the MS/MS spectra of TMT-labeled peptides. Peptide quantification was analyzed using the WSPP model, which uses raw quantifications as input data and computes the protein log2-fold changes for each individual with respect to the average of the values of the two reference internal standard samples. In this model protein log2-ratios are expressed as standardized variables in units of standard deviation according to their estimated variances (*Zq* values).

The mass spectrometry proteomics data have been deposited to the ProteomeXchange Consortium via the PRIDE [27] partner repository with the dataset identifier PXD034045.

### 2.8. Peptide Synthesis

Light and heavy versions of the selected peptides were synthetized using standard F-moc chemistry. We used ^13^C and ^15^N lysine and arginine for SIL heavy peptides, resulting in an 8- and 10-Da mass shift, respectively, compared with their light counterparts. For purification of synthetic peptides, Cys residues were blocked with 50 mM iodoacetamide for 1 h at 37 °C and separated on a C18 reversed phase column with a 0–65% ACN gradient at 2 mL/min (JASCO Pu-2089 Plus pump coupled to a JASCO UV-2077 detector, Tokyo, Japan). The UV detector was set at 214 and 280 nm to monitor eluting peptides. Main chromatographic peaks were collected and re-analyzed for purity assessment by high resolution analytical LC on a Scharlau C18 column (5-μm particle size, 2 mm I.D. × 25 cm) using a 40-min 0–70% ACN gradient at 250 µL/min at 40 °C (Ultimate 3000 HPLC). The minimal purity of all peptides was 90%. Molecular weight and peptide purity was further assessed by matrix assisted laser desorption ionization time of flight (MALDI TOF/TOF) analysis (SCIEX 4800, Framingham, MA, USA).

### 2.9. SRM Analysis

Five microliters of serum sample were denatured in 395 μL 6M urea in 100 mM TRIS pH 7.8 for 30 min at room temperature. Fifty microliters of the mixture was reduced by the addition of 1.5 µL of 200 mM DTT during 30 min at 37 °C and alkylated by subsequent addition of 6 µL of 200 mM iodoacetamide during 30 min at room temperature in the dark. Unreacted iodoacetamide was consumed by adding 6 µL of 200 mM DTT over 30 min at room temperature. The samples were diluted with 250 µL of 50 mM ammonium bicarbonate to a final concentration of less than 1M urea. Denatured and reduced proteins were then digested by the addition of 15 µL of 0.1 μg/μL trypsin (Gold Trypsin, Promega, Madison, WI, USA) and incubated overnight at 37 °C. The reaction was stopped the next day by adding concentrated formic acid (FA) (Sigma). The digest was evaporated in the vacuum concentrator and resuspended in 90 µL of 2% ACN, 0.1% FA. The resulting peptide concentration was measured with the Qubit Assay Protein kit on a Qubit 3.0 fluorimeter (ThermoFisher, Waltham, MA, USA) following the manufacturer’s instructions.

The development of the SRM method for the quantification of the proteins of interest as well as the quantification of the samples were carried out using the Skyline software (Appendix A) (MacCoss Lab Software, Seattle, WA, version 20.2.0.343). Only proteotypic peptides were selected for each protein prioritizing those used in the TMT experiment and those from the SRMAtlas repository (http://www.srmatlas.org/, accessed on 18 October 2021). The selected peptides were between 7–25 amino acids in length. A minimum of three transitions were selected for each peptide based on the intensity of the fragment ions and the MS/MS spectra of the human library. SRM analyses were performed on a triple quadrupole/linear ion trap hybrid mass spectrometer (6500QTRAP+, Sciex, Foster City, CA, USA) coupled to a nano/micro-HPLC (Eksigent LC425, SCIEX, Framingham, MA, USA). The preconcentration and desalting of the samples was performed online using a C18 guard column (Luna^®^ 0.3 mm id, 20 mm, 5 µm particle size, Phenomenex, CA, USA) at 10 µL/min for 3 min. Approximately 2 µg of digest was injected according to Qubit’s peptide quantification. Peptide separation was performed using a C18 column (Luna^®^ Omega Polar 0.3 mm id, 150 mm, 3 µm particle size, Phenomenex, CA, USA), at 5 µL/min and an oven temperature of 40 °C. The elution gradient was 5% to 35% ACN (0.1% FA) in 30 min. The mass spectrometer is interfaced with an ESI source (Turbo V ™) using a 25 µm ID hybrid electrode and is operated in positive mode. The source parameters were: capillary voltage 5000 V, potential declustering 85 V, curtain gas and Gas1 (nitrogen) 25 psi, gas2 (nitrogen) 15 psi, temperature 150 °C.

The retention times of the synthetic isotopically labeled peptides (SIL peptides) were used to generate a final scheduled SRM method. The final scheduled SRM method was tested on a pool of SIL peptides spiked into control samples and it was confirmed that all spiked-in peptides could be identified by the method. For each peptide, transitions were recorded in scheduled MRM mode with a time window of 120s. The collision energies were calculated automatically using the collision energy equations built into the Skyline software. In order to confirm the identity of the peptides, an MRM-initiated detection and sequencing experiment (MIDAS) was performed for each peptide. The mass spectrometer was programmed to switch from MRM scanning mode to enhanced product ion (EPI) mode when an MRM transition signal exceeded 100,000 counts.

For each protein, a master mix containing the heavy peptides of each protein was prepared at 10 pmol/μL. Then, for each protein, a different concentration of SIL peptides was spiked on samples depending on the intensity of endogenous peptides in samples so that the light to heavy ratio was approximately one. Final concentrations of SIL peptides spiked in the samples were between 2–100 fmol/μL and protein concentration was estimated with the light to heavy ratio.

Peptide and protein quantification was analyzed using the same statistical model as for TMT analysis. Briefly, peptide concentrations were used as input data and protein log2-fold changes were computed for each individual with respect to the average of peptide concentration in all individuals analyzed.

### 2.10. Statistical Analysis

Logistic regression models were tested using SPSS software (IBM, Armonk, New York, NY, USA). Statistical significance of changes (*p*-value) was calculated using two-tailed Student’s *t*-test for binary comparisons, or one-way ANOVA for multiple comparisons.

## 3. Results

### 3.1. Serum Proteomic Analysis of the Discovery Cohort

In this work, we made a systematic analysis of the serum proteome from 72 patients at different ages (<60, 60–80 and >80) suffering from mild to severe COVID-19 who are representative from the Spanish population (Figure 1). This discovery cohort included four separate groups: Non-Hospitalized, Hospitalized, ICU and EXITUS, and their clinical characteristics are depicted in Table 1A. The alterations in the serum proteome were validated by targeted proteomics in a second cohort containing 84 individuals (Figure 1), which contained an additional group with discharged individuals, and whose characteristics are listed in Table 1B.

The discovery cohort containing 72 COVID-19 serum samples was analyzed by hypothesis free quantitative LC-MS using TMT multiplexed isobaric labeling and without depletion of the most abundant proteins to preserve the integrity of the serum proteome. Peptide fractionation was used instead (Figure 1), an approach that improved peptide and protein yield and that has been demonstrated to be a reproducible method in previous studies [28]. Protein quantification was performed on the basis of the WSPP model using the SanXoT package, as we have done in previous serum proteomics studies [29,30,31,32,33]. We observed that 90% of the proteins had a coefficient of variation <20% which is the commonly used cutoff for diagnostic assays. Besides, we were able to quantify 67 out of 109 of the FDA-approved biomarkers (FDA-NIH: Biomarker-Working-Group, 2016) and only one of them showed a CV higher than 20. We were able to quantify a mean of 1006 proteins per sample (Appendix A). Among them, 53 proteins were differentially regulated in at least one of the intergroup comparisons (Figure 2). The cluster of up-regulated proteins were implicated in acute-phase response and inflammation, blood coagulation, lung and kidney damage, immune response and complement cascade. Down-regulated proteins were mainly implicated in lipid metabolism, oxidative-stress and cell-adhesion processes (Figure 2).

Among the proteins that showed the highest statistical significance in the comparison between Non-Hospitalized and EXITUS patients, we selected six proteins that were representative of the main processes altered with the disease (SERPINA3, FGA, PON1, AFM, APOA2 and TTR) and analyzed their overall association with disease severity using a logistic regression model. We observed that the prediction of the logistic model reflected the severity of disease, yielding a gradual increase of proteome changes from Non-Hospitalized to EXITUS patients (Figure 3). Moreover, our data suggest that proteome changes increased with age within the hospitalized group of patients (Figure 3).

### 3.2. Targeted Analysis in the Validation Cohort

To study whether these changes were reproduced in a different set of individuals, we selected 18 proteins that were representative of the biological processes altered in the discovery cohort (Appendix A); we also added HPT, CFB and C5 to the list since these proteins have been described to change by SARS-CoV-2 infection in other cohorts (Appendix A). These proteins were quantified by SRM-targeted proteomics in an independent cohort of 84 patients that included the same groups of patients as the discovery cohort and 21 additional individuals that successfully recovered from SARS-CoV-2 infection (validation cohort). Protein quantification was based on two to four peptides and data were only accepted when there was a good correlation among the abundance ratio of the monitored peptides with the spectral library (Appendix A). The majority of the monitored proteins in the validation cohort displayed abundance changes that reproduced the observations made in the discovery cohort, with the exception of FBLN1, FGA, SERPINC1 and IGFALS that did not change in the SRM analysis but were found altered in the TMT analysis; and HPT that although was not altered by TMT was found to be up-regulated in SRM with increasing the severity of the disease, in concordance with other studies (Figure 4A). The level of all these proteins returned to normal values 100 days after patient discharge, further supporting their association with COVID-19 (Figure 4A). Using a logistic regression model with the same set of six proteins selected in the discovery cohort we observed that the overall changes in the serum levels of these proteins also reflected the gradation of the disease (Figure 4B).

### 3.3. Comparison with Other Studies

Despite biological heterogeneity and technical variables associated to the analytical pipelines used in different laboratories that usually hinder the generation of an integrated outcome, we have found a significant concordance with previous studies. The degree of overlapping of our protein panel with previous MS-based studies ranges from 10% to 70% while it remains below 10% when it is compared to the proteins reported from studies using other platforms such as OLINK (Appendix A), likely due to the a priori selection of the target protein set to be analyzed by the proximity extension assay. Of note, the studies including more than 45% of the differential proteins reported in our study [12,34,35,36] have been done using plasma or serum samples and three different MS platforms: orbitrap, TIMS TOF and 6600 QTOF. Moreover, the observation frequency of the protein alterations across independent studies might be considered as a reference of its reliability and its association with COVID-19. Of the regulated protein panel, thirty-four proteins have been also reported in five or more studies, twenty-four in two to five, and four were reported only in one study from other authors [10,11,12,13,14,34,35,36,37,38,39,40,41,42,43,44,45,46]; specifically, CD14, CRP, ITIH1, ITIH3, LBP, LRG1, SERPINA3, VWF, C9, ApoA1 and GSN have been associated with COVID-19 severity in at least eight independent studies (Appendix A).

## 4. Discussion

The COVID-19 pandemic represents a global challenge from health, societal and economics perspectives that has evidenced the need to make efficient resources available to combat its deleterious effects and to prepare preventive strategies for future pandemics. The worldwide impact and the different idiosyncrasies of different geographical regions make it necessary to conduct studies involving different populations to identify specific factors that might lead to more selective and efficient clinical strategies. Proteomics has been extensively used to gain insights on the molecular bases of SARS-CoV-2 infection from different perspectives, including the monitorization of serum proteome variations as a readout of the host response that opens new perspectives for a more efficient management of COVID-19 patients. However, to date only one work has been published studying the impact of SARS-CoV-2 infection in the serum proteome in a Spain cohort [31].

In this study we performed a hypothesis-free proteomics analysis of alterations in serum proteins in a cohort of patients from Spain. We detected a set of serum proteins whose abundance was altered with increasing disease severity, and which also might correlate with the age of individuals in the group of hospitalized patients. The majority of these changes were confirmed in a validation cohort by targeted proteomics. Besides, a systematic comparison with other published works showed that most of the serum protein changes are in good concordance with previous studies, confirming that SARS-CoV-2 infection is consistently associated with specific serum protein alterations.

Among the differentially regulated proteins that mediate the acute-phase response found in our study, CRP, SAA1 and ALB alterations likely result from macrophage activation through SARS-CoV-2 binding to the ACE receptor and activation of an IL6 mediated response [47]. IL6 is an early driver of the host cell response to SARS-CoV-2 infection that involves the regulation of key inflammatory genes, including ITIH proteins. ITIH 1–4 are a family of proteins that form a complex with the bikunin protein through a chondroitin sulfate bridge (inter-α-trypsin inhibitor and pre-a-inhibitor) and play a myriad of biological functions depending on their tissue localization and macromolecular assembly configuration [48]. Differential expression of ITIHs has been associated with the severity and progression of COVID-19 in different proteome wide studies of serum and plasma samples [10,34,35,36,37,38,43,45,46]. Similarly, to other studies, we observed a systematic ITH1 and 2 decrease and ITIH3 and 4 increase, paralleling the severity of COVID-19 patients. Interestingly, these changes showed a good correlation with age in the group of hospitalized patients, suggesting ITIHs as prognostics indicators since ageing is considered a risk factor for severe COVID-19.

Despite the detected changes on several regulators of the complement cascade, we could not gather enough evidence supporting complement or alternate pathway activation in our populations. Hence, we have no bases to recommend complement-based therapeutics for the management of COVID-19 patients.

In the other hand, CD14, LBP and LRG1 might indicate the progression of lung damage in COVID-19 patients. CD14 and LBP are mediators of lung inflammation that are produced in response to infectious processes and LRG1 is a hallmark of risk for lung fibrosis assessment [49]. Based on this evidence, their role in the progression of the respiratory syndrome associated with COVID-19 has been postulated as well as their potential as therapeutic targets to prevent this complication in the more severe cases. Other indicators of tissue damage are GSN, ACTB and CST3. Circulating GSN is part of the extracellular actin scavenger system (EASS) as it has the ability to bind actin filaments released from necrotic cells. Its depletion has been observed in diverse states of inflammation associated with tissue injury, including adult respiratory distress syndrome (ARDS), sepsis, myocardial infarction, hepatitis and rheumatoid arthritis, among others [50,51,52,53] and is commonly associated with poor prognosis. CST3 is a low molecular weight protein with a central role in connective tissue remodeling through inhibition of extracellular cysteine proteinases [54]. It has been described that patients with increased CST3 levels having mild-to-moderate kidney disfunction had a higher risk of major adverse coronary events; indeed, this protein has been used as a marker for estimated glomerular filtration rate [55] and is considered as a poor prognosis predictor in patients with chronic kidney disease, which is characterized by a strong immune and inflammatory component [56,57]. Moreover, acute kidney injury is common among critically ill patients with COVID-19, affecting approximately 20–40% of patients admitted to intensive care [58]. Virus particles were reported to be present in renal endothelial cells, indicating viraemia as a possible cause of endothelial damage in the kidney and a probable contributor to AKI, an effect that may be partially mediated by upregulation of CST3 [59].

In addition to inflammation and tissular damage, our data support activation of coagulation although no alterations on platelet degranulation were found. The latter might be a cohort-dependent effect since it has been observed in some studies [60] but not in others [46], including this one. Coagulation activation could result in thrombotic manifestations of SARS-CoV-2 infection and may suggest that anticoagulant prophylaxis might be beneficial to prevent complications mainly associated with severe COVID-19 cases, as we have recently reported [31]. According to this hypothesis, it has been shown that the elevation of D-dimer is an early predictor of thrombosis in COVID-19 cases, although the benefit of an early administration of anticoagulants must be tested in specifically designed randomized clinical trials [61].

Finally, similarly to the results from other cohorts, we detect a marked tendency to downregulation of apolipoproteins, paralleling the increase of COVID-19 severity. APOA1, APOA2 and APOL1 are components of HDL, mainly produced in the liver, which are recognized modulators of immunity and inflammation [62]. Furthermore, it is worth noting that HDL decrease is a risk factor for cardiovascular disease and therefore, low HDL levels may indicate a higher risk of cardiovascular complications [63] and kidney damage [64] for COVID-19 patients.

In conclusion, this work supports the notion that SARS-CoV-2 infection produces a clear impact on the serum proteome, reflecting the alteration of several biological processes that improve our understanding of COVID-19 disease and provide opportunities to monitor its severity. Our results, obtained using two cohorts from Spain, reinforce the results obtained in other cohorts. Despite the different platforms used, the specific study design, the type of sample used (serum or plasma), the different sample processing and data analysis procedures, the high degree of inter-laboratory overlapping and the concordance of the functional processes affected across proteomics studies provides a solid background for future developments for an improved clinical management of COVID-19 patients.

## Figures and Tables

**Figure 1 biomedicines-10-01690-f001:**
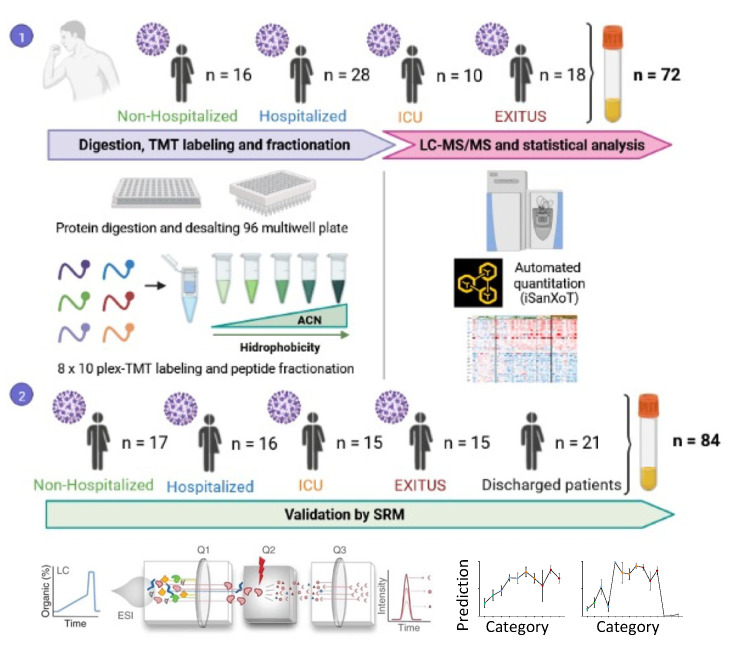
Experimental workflow Seventy-two serum samples (discovery cohort) from patients at different ages suffering from mild to severe COVID-19 were subjected to digestion, TMT-labeling, peptide fractionation, LC-MS/MS and statistical analysis with SanXoT package. To confirm our previous observations 18 proteins were selected for targeted validation by SRM in an independent cohort of 84 patients that included 21 individuals that successfully recovered from SARS-CoV-2 infection.

**Figure 2 biomedicines-10-01690-f002:**
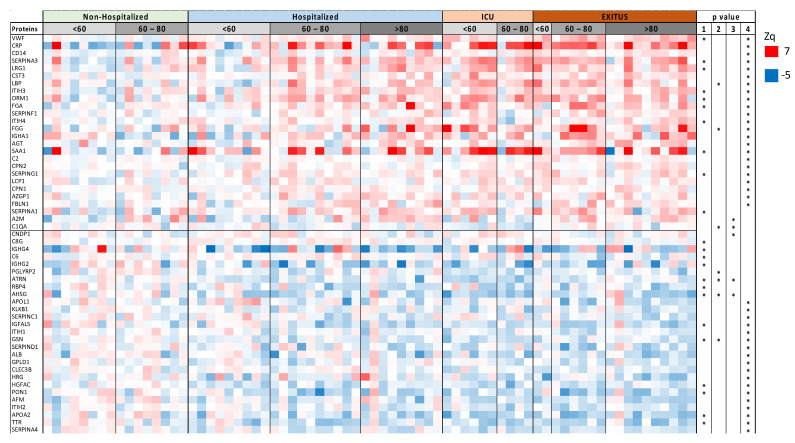
Proteins differentially regulated with disease severity. The heatmap shows protein abundance changes (expressed as Zq, or standardized log2-ratios) normalized by the average values of the non-hospitalized patient group. Statistical significance of changes (*p*-value) is calculated using two-tailed Student’s *t*-test. Proteins (quantified with more than one peptide and in the 80% in the individuals) whose abundance is significantly (*p*-value < 0.05) increased (red) or decreased (blue) at least in one comparison are shown (1 = Non-Hospitalized vs. Hospitalized, 2 = Hospitalized vs. ICU, 3 = ICU vs. EXITUS, 4 = Non-Hospitalized vs. EXITUS).

**Figure 3 biomedicines-10-01690-f003:**
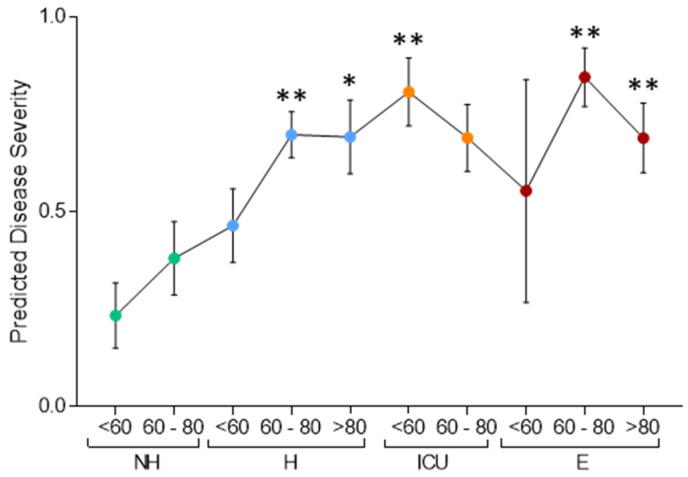
Logistic regression model using selected proteins that correlate with disease progression and severity in the discovery cohort. The panel was constructed with SERPINA3, FGA, PON1, AFM, APOA2 and TTR. The model was trained by logistic regression comparing the NH and E groups. Data are mean ± SEM of the prediction of the model per each individual. Statistical significance (*p*-value) is calculated using one-way ANOVA, (* *p* < 0.05, ** *p* < 0.01) and all the comparisons refer to <60 years-old non-hospitalized patients.

**Figure 4 biomedicines-10-01690-f004:**
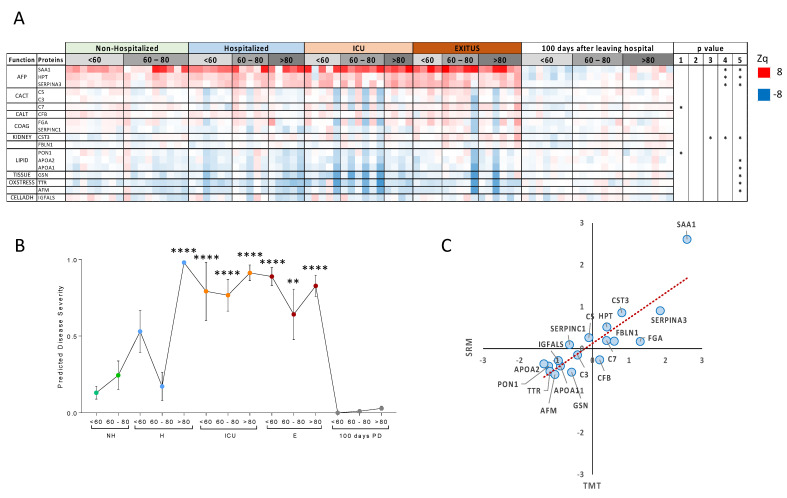
Proteins monitored by SRM. (**A**) The heatmap shows protein abundance changes (Zq) normalized by the average values of the 100 days after leaving hospital patient group. Up-regulated proteins are shown in red and down-regulated proteins in blue. Statistical significance of changes (*p*-value) is calculated using two-tailed Student’s *t*-test (1 = Non-Hospitalized vs. Hospitalized, 2 = Hospitalized vs. ICU, 3 = ICU vs. EXITUS, 4 = Non-Hospitalized vs. EXITUS, 5 = Exitus vs. 100 days after patient discharged). (**B**) Prediction of disease severity using a logistic regression model. The model was trained using the same proteins as in Figure 3. (**C**) Protein quantification from TMT and SRM experiments. Protein quantification is represented as the average difference of Zq values between Non-Hospitalized and EXITUS patients. Statistical significance (*p*-value) is calculated using one-way ANOVA (** *p* < 0.01, **** *p* < 0.0001 and all the comparisons refer to <60 years-old non-hospitalized patients.

**Table 1 biomedicines-10-01690-t001:** Characteristics of discovery and validation cohorts.

	**Non-Hospitalized**				**Hospitalized**					**ICU**		**EXITUS**							
**A. Discovery cohort**	**<60**		**60–80**		**<60**	**60–80**		**>80**		**<60**	**60–80**	**<60**			**60–80**		**>80**		
	**(n = 8)**		**(n = 8)**		**(n = 9)**	**(n = 10)**		**(n = 9)**		**(n = 6)**	**(n = 4)**	**(n = 2)**			**(n = 6)**		**(n = 10)**		
Age (years), mean (SD)	46 ± 7		68 ± 5		37 ± 4	63 ± 2		83 ± 2		49 ± 6	66 ± 4	50 ± 8			69 ± 4		85 ± 3		
Male, No. (%)	3 (38%)		4 (50%)		5 (56%)	7 (70%)		3 (33%)		5 (83%)	3 (75%)	2 (100%)			3 (50%)		3 (30%)		
Days between symptoms onset and plasma extraction, mean (SD)	7 ± 2		9 ± 2		9 ± 2	9 ± 2		7 ± 1		9 ± 2	8 ± 1	9			8 ± 2		9 ± 3		
Comorbidity, No. (%)	5 (63%)		6 (75%)		2 (22%)	7 (70%)		9 (100%)		5 (83%)	3 (75%)	2 (100%)			6 (100%)		9 (90%)		
Pharmacotherapy, No. (%)	6 (75%)		6 (75%)		4 (44%)	8 (80%)		9 (100%)		6 (100%)	4 (100%)	0			5 (83%)		8 (80%)		
	**Non-Hospitalized**				**Hospitalized**					**ICU**		**EXITUS**				**Discharged (100 days)**			
**B. Validation cohort**	**<60**	**60–80**		**>80**	**<60**		**60–80**		**>80**	**<60**	**60–80**	**<60**	**60–80**	**>80**		**<60**		**60–80**	**>80**
	**(n = 6)**	**(n = 8)**		**(n = 3)**	**(n = 6)**		**(n = 5)**		**(n = 5)**	**(n = 5)**	**(n = 10)**	**(n = 4)**	**(n = 5)**	**(n = 6)**		**(n = 7)**		**(n = 7)**	**(n = 7)**
Age (years), mean (SD)	44 ± 9	68 ± 5		88 ± 7	43± 9		68 ± 6		86 ± 3	39 ± 7	72 ± 5	49 ± 9	70 ± 6	90 ± 4		53 ± 5		72 ± 6	85 ± 4
Male, No. (%)	1	4		2	3		3		2	4	9	3	1	1		7		4	2
	(17%)	(50%)		(67%)	(50%)		(60%)		(40%)	(80%)	(90%)	(75%)	(20%)	(17%)		(100%)		(57%)	(29%)
Days between symptoms onset and plasma extraction, mean (SD)	6 ± 4	8 ± 3		5 ± 2	7 ± 4		6 ± 4		6 ± 2	6 ± 3	8 ± 2	9 ± 2	6 ± 4	5 ± 3		170 ± 52		150 ± 55	166 ± 52
Comorbidity, No. (%)	6	7		3	4		4		5	5	9	3	5	6		5		7	7
	(100%)	(87%)		(100%)	(67%)		80%)		(100%)	(100%)	(90%)	(75%)	(100%)	(100%)		(71%)		(100%)	(100%)
Pharmacotherapy, No. (%)	5	4		3	3		4		4	5	8	2	4	4		5		7	7
	(83%)	(50%)		(100%)	(50%)		(80%)		(80%)	(100%)	-0,8	(50%)	(80%)	(67%)		(71%)		(100%)	(100%)
Age (years), mean (SD)	44 ± 9	68 ± 5		88 ± 7	43± 9		68 ± 6		86 ± 3	39 ± 7	72 ± 5	49 ± 9	70 ± 6	90 ± 4		53 ± 5		72 ± 6	85 ± 4

## Data Availability

Proteomics data have been uploaded into PRIDE repository and can be accessed with the reference PXD034045.

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
