# Peer review of "Mapping the Serum Proteome of COVID-19 Patients; Guidance for Severity Assessment"

_biomedicines, 2022, doi:10.3390/biomedicines10071690_

Round 1

Reviewer 1 Report

Nuñez et al. present a sound serum proteome mapping for COVID-19 cohorts from Spain; the methodology seems quite convincing, but the data representation and tables need some attention. The relatively good coverage of the FDA-approved biomarkers within a reasonable CV-range is a good argument for this study. Subsequently, the verification of the identified biomarkers within a SRM assay is straight-forward; I would have loved to also see, whether the discovery cohort (1) could also benefit from the SRM assays in terms of accuracy (maybe as a supplemental data, for a subset of markers/files? I am especially thinking of figure 3).

An annoying fact: I could not access the raw PRIDE data submission – the authors need to supply reviewer access for the time of review at least (PRIDE supplies reviewer access IDs upon upload).

There are also “Spanish special characters” present at some places within the English text. In general, why were the discovery and verification cohort processed differently according to the methods section? I would also like to see a general graphics panel for the identified proteins in terms of rank plot, MW etc. to give a general overview of the discovery dataset (as a suppl. figure maybe?) and a comparison of the quantification for the selected biomarker panel by SRM and TMT (grabbing up the statements in L329 onwards).

Detailed feedback for the paper follows:

·         - L23: … vaccines3 ß why this superscript?

·          - L100: right parenthesis missing

·         - L104: formatting not fitting to general text; Why are the authors denaturing only at 56 °C? Especially afterwards the authors mention boiling with DTT for 5 minutes – is this boiling at 56 °C or at 95-100°C as I would understand it? Needs clarification.

·         - L122: How is reference channel build up, are all serum samples pooled or just some specific ones?

·         - L137: Why are the authors choosing this fractionation scheme? Thermo’s general fractionation scheme includes also lower ACN-% elution steps, which generally yield hydrophilic peptides. Why do the authors omit those?

·         - L147: Is the gradient really running straight from A to 100%B over the whole 338 min.?

·         - L162: 800ppm precursor mass tolerance for a QE HF?

·         - Figure 1: Can resolution be improved?

·         - Table 1: It would be nice to include the comorbidity “smoking” as a separate entity.

·         - Figure 2: The legend for z-score scale is missing. Do not color the age groups with green, as green has been already used for the Non-Hospitalized group – maybe select some gray shades?

·         - L289: What was the rationale for the exact selection of those markers?

·         - Figure 3: The significance asterisks are partially not clearly attributable (does this belong to <60 or 60-80 ICU group, for example).

·         - Figure 4: why are for the H/E groups the 60-80 groups less severely diseased? Why is there a strong decline is disease severity for the markers in those age groups? Please discuss.

·         - L390: Why are some ITIHs down- and some up-regulated? Is there a physiological background for this differential regulation, especially in the context of COVID-19?

·         - L427: COVID-19

·         - Supplemental data:

o   Supp. Table 1 should feature a small legend regarding the table headers (NH, H…); in addition, this table lacks the gene, peptides, unique peptides and molecular weight information. The data is z-score normalized? Then it should be stated within the table itself.

o   Supp. Table 2 should also feature the fragment masses for each transition.

o   In Supp Table 3 there are some Spanish comments (rightmost part), those should be supplied in English. I would like to know the referenced publications and studies directly within the Excel file without the need to look them up; please supply a separate sheet with study code & full reference in the Excel file. Maybe it would be nice to have a consistent study code (Author-YEAR) and a heading with a merged table cell stating Olink vs. the rest.

o   Supp Fig 1 should definitively be separate and not supplied within the Excel-File as the readability is rather bad; please supply as a separate high-resolution figure.

Author Response

Reviewer 1

Nuñez et al. present a sound serum proteome mapping for COVID-19 cohorts from Spain; the methodology seems quite convincing, but the data representation and tables need some attention. The relatively good coverage of the FDA-approved biomarkers within a reasonable CV-range is a good argument for this study. Subsequently, the verification of the identified biomarkers within a SRM assay is straight-forward; I would have loved to also see, whether the discovery cohort (1) could also benefit from the SRM assays in terms of accuracy (maybe as a supplemental data, for a subset of markers/files? I am especially thinking of figure 3).

R.- We fully appreciate the positive comments by the reviewer. Concerning the possibility of doing SRM analysis in the discovery cohort, this is a really interesting idea; unfortunately, the editor has only granted us 10 days to answer reviewers’ questions providing a revised manuscript. It would be impossible to do the SRM analysis in this timeframe.

An annoying fact: I could not access the raw PRIDE data submission – the authors need to supply reviewer access for the time of review at least (PRIDE supplies reviewer access IDs upon upload).

R.-We apologize for the inconvenience. When we submitted the manuscript, we checked that the data was accessible with the identifier we supplied for the reviewers. We do not understand why it is not accessible now. We have contacted PRIDE for a new reviewer access number that should be valid throughout the time of review. The new access provided by PRIDE is the following

username: reviewer_pxd034045@ebi.ac.uk and password: AglkMn2T

There are also “Spanish special characters” present at some places within the English text.

R.- We thank the reviewer for this observation. All Spanish characters have been removed.

In general, why were the discovery and verification cohort processed differently according to the methods section?

R.- We appreciate the reviewer’s comment and apologize for the misunderstanding. The samples of the discovery and verification cohort were processed in different laboratories, each one with their optimized virus inactivation and digestion protocols that are widely accepted.  We acknowledge that we did not give enough details about how data from SRM experiments were processed. The discovery and validation cohorts were processed using the same statistical model, and the quantitative protein values have the same meaning. We now explain how SRM data analysis was performed in the revised version of the manuscript (Lines 255-258).

I would also like to see a general graphics panel for the identified proteins in terms of rank plot, MW etc. to give a general overview of the discovery dataset (as a suppl. figure maybe?)

R.- A rank plot showing the distribution of protein abundances has been added as Supp. Figure 2. The MW have been added as a new column to Supp. Table 1.

and a comparison of the quantification for the selected biomarker panel by SRM and TMT (grabbing up the statements in L329 onwards).

R.- Thank you for your suggestion; this is a very good idea. We have included in the manuscript a new Figure 4C where we compare the quantification results obtained for the selected proteins by SRM and TMT analysis. The new figure shows the good agreement between the two cohorts.

Detailed feedback for the paper follows:

 L23: … vaccines3  why this superscript?

R.- This was an old reference and has been removed.

L100: right parenthesis missing.

R.- Corrected, many thanks

L104: formatting not fitting to general text; Why are the authors denaturing only at 56 °C? Especially afterwards the authors mention boiling with DTT for 5 minutes – is this boiling at 56 °C or at 95-100°C as I would understand it? Needs clarification.

R.- Thank you for your comment. Samples were previously heated in the presence of SDS at 56°C during 30 minutes for virus inactivation, and later protein denaturation was performed by adding DTT and boiling at 100°C. The sentence has been modified in the text for a better understanding (Lines 105-106).

L122: How is reference channel build up, are all serum samples pooled or just some specific ones?

R.- The reference channel is created by pooling up serum samples from all the individuals of the cohort in each TMT batch. This sentence has been changed in the text to have a better understanding.

L137: Why are the authors choosing this fractionation scheme? Thermo’s general fractionation scheme includes also lower ACN-% elution steps, which generally yield hydrophilic peptides. Why do the authors omit those?

R.- This method has been optimized for plasma samples and in our hands is the one that gives the best performance in terms of identified proteins. In fact, we consistently identify a considerably lower number of peptides in the lower ACN fractions than in the rest of fractions, so that we do not expect to increase performance by further fractionating these low ACN fractions.

L147: Is the gradient really running straight from A to 100%B over the whole 338 min.?

R.- We thank the reviewer for this important observation. We apologize for the mistake; there was an error in the sentence. The gradient ends at 100%ACN but it is not a linear gradient. We used an acetonitrile gradient consisting of 0-21% buffer B (100% acetonitrile, 0.5% formic acid) for 300 min and 21–90% for 5 min. This sentence has been corrected in the revised version of the manuscript (Lines 153-154).

L162: 800ppm precursor mass tolerance for a QE HF?

R.- Precursor mass tolerance for the database search is set at 800ppm, but later, as we say in the text, we use an additional filter for precursor mass tolerance of 15ppm at the time of calculating the false discovery rate (FDR). This postfiltering approach avoids problems related to direct searches with low precursor mass tolerances and at the same time decreases FDR, as it has been previously published in our group and referenced in the text (Bonzón-Kulichenko et al., Journal of Proteome Research 2015).

Figure 1: Can resolution be improved?

R.- Figure 1 resolution has been improved and watermark has been removed.

Table 1: It would be nice to include the comorbidity “smoking” as a separate entity.

R.- Thank you for your comment. Unfortunately, the biobank do not have this information for all patients and therefore we did not included it in the table.

Figure 2: The legend for z-score scale is missing. Do not color the age groups with green, as green has been already used for the Non-Hospitalized group – maybe select some gray shades?

R.- Thank you for your comments. Z-score scale has been added and age groups colors have been modified. These changes have been extended to Figure 4 and Supp. Figure 1

L289: What was the rationale for the exact selection of those markers?

R.- The selected markers are the proteins representative of the main processes altered and the ones that had the highest statistical significance in the comparison of non-hospitalized patients versus EXITUS patients. This sentence has been modified in the revised version of the manuscript to have a better understanding (Lines 308-309).

Figure 3: The significance asterisks are partially not clearly attributable (does this belong to <60 or 60-80 ICU group, for example).

R.- We are sorry that asteriks must have been misplaced at the time of generating the PDF file during the submission step. We will recheck them in the next submission.

Figure 4: why are for the H/E groups the 60-80 groups less severely diseased? Why is there a strong decline is disease severity for the markers in those age groups? Please discuss.

R.- We agree with the reviewer that the reduced severity prediction in these specific cases was not expected. We do not have a conclusive explanation for this behavior based on biological or clinical parameters. However, although patients were selected according to homogeneous clinical parameters it might be possible that the reduced number of cases of each group may limit the predictive capacity of the model in particular cases. Therefore, conclusions based on the overall trend instead on a point-by-point observations would be more accurate.

L390: Why are some ITIHs down- and some up-regulated? Is there a physiological background for this differential regulation, especially in the context of COVID-19?

R.- The exact role of ITIH proteins in serum is not fully understood. They have been associated with extracellular matrix homeostasis and downregulation of ITIH1 has been reported in patients with sepsis, correlating with high mortality rates (Opal SM et al Crit Care Med 2007. DOI:10.1907/01CC). Interestingly our data are in good agreement with those previously published by Vollmy et al (Vollmy et alLife Sciences Alliance 2021. DOI.org/1026508/Isa2021101099) who found downregulation of ITIH1 and 2 and upregulation of ITIH3 and 4 in no survivors COVID-19 patients respect to survivors. While the functional understanding of these alterations needs further investigation the four related ITIH proteins would provide a panel for monitoring disease outcome in different pathogen-caused diseases, COVID-19 among them.

Supplemental data:

Supp. Table 1 should feature a small legend regarding the table headers (NH, H…); in addition, this table lacks the gene, peptides, unique peptides and molecular weight information. The data is z-score normalized? Then it should be stated within the table itself.

R.-Thank you for your comments. A small legend has been added to Supp. Table 1 providing information about table headers and protein quantification values (Zq). In addition, two new columns with Gene and MW data have been added to this table. A new Supp. Table 4 has been also created including peptides identified in each protein (including unique peptides and peptides that were assigned to the razor protein).

Supp. Table 2 should also feature the fragment masses for each transition.

R.- Fragment masses have been included in the new version of supplementary table 2, as suggested by the reviewer.

In Supp Table 3 there are some Spanish comments (rightmost part), those should be supplied in English. I would like to know the referenced publications and studies directly within the Excel file without the need to look them up; please supply a separate sheet with study code & full reference in the Excel file. Maybe it would be nice to have a consistent study code (Author-YEAR) and a heading with a merged table cell stating Olink vs. the rest.

R.- We acknowledge the reviewer comments. Table 3 has been now modified as suggested. Column headings have been modified, the references have been included, Spanish characters have been removed and OLINK studies have been indicated.

Supp Fig 1 should definitively be separate and not supplied within the Excel-File as the readability is rather bad; please supply as a separate high-resolution figure.

R.- We thank the reviewer for this comment. However, only one file for supplementary information is allowed in the submission format.

Reviewer 2 Report

The present manuscript presents valuable and interestingly information regarding serum proteome of COVI-19 patients, especially relating the proteomics output with disease severity and according to age intervals. The methods are well described and compared with data obtained from other authors. I have only the following comments:

-      Despite the title highlighting the origin from patients (Spain), the manuscript does not compare the results according to other geographies as obtained from other authors. Of course, geography could be relevant, but as it can be relevant other variables, that are not highlighted in the title. For that, I strongly suggest removing from the title and abstracts the geography highlight.

-      Please avoid adding references in the abstract

-      The first time that and abbreviation is referred to, it should be proceeded by its description in extension, as was the case of “LC-ESI-MS/MS”, DTT, etc

-      Table legend are usually on top of table

-      Line 376, it is indicated “confirmed in a discovery cohort by targeted proteomics”, but was not the validation cohort?

Author Response

The present manuscript presents valuable and interestingly information regarding serum proteome of COVI-19 patients, especially relating the proteomics output with disease severity and according to age intervals. The methods are well described and compared with data obtained from other authors. I have only the following comments:

Despite the title highlighting the origin from patients (Spain), the manuscript does not compare the results according to other geographies as obtained from other authors. Of course, geography could be relevant, but as it can be relevant other variables, that are not highlighted in the title. For that, I strongly suggest removing from the title and abstracts the geography highlight.

R.- We thank the reviewer for this comment. The geographical highlight has been removed from the title and abstract.

Please avoid adding references in the abstract

R.- Done as suggested, many thanks.

The first time that and abbreviation is referred to, it should be proceeded by its description in extension, as was the case of “LC-ESI-MS/MS”, DTT, etc

R.- Done as indicated. Many thanks

Table legend are usually on top of table

R.- The title of tables has been placed on top, as indicated by the reviewer.

Line 376, it is indicated “confirmed in a discovery cohort by targeted proteomics”, but was not the validation cohort?

R.- We thank the reviewer for this observation. Indeed, we wanted to say ‘confirmed in a validation cohort by targeted proteomics’. We apologize for the mistake, this sentence has been modified in the text.

Reviewer 3 Report

The authours have completed a fairly low depth analysis of the serum proteome of COVID patients. The results identify a number of modestly significant differences in a small group of proteins. Many of these have been identified by other groups using related but not  identical LC MS methods.  A selected subset of candidates were analysed by reaction monitoring with corroboration of some but not all candidates. 

The results are interesting and may be of some value to the research community. Although none of the candidates appears to be likely useful as disease status markers, in conjunction with the results of other studies they may be useful for characterising disease process. 

Given the lack of data on non admitted patients >80,  I think that the authours should temper their statements to indicate that there may an age related difference rather than there is.

Author Response

The authours have completed a fairly low depth analysis of the serum proteome of COVID patients. The results identify a number of modestly significant differences in a small group of proteins. Many of these have been identified by other groups using related but not  identical LC MS methods.  A selected subset of candidates were analysed by reaction monitoring with corroboration of some but not all candidates. 

R.- We thank the reviewer for the careful revision and relevant discussion. As mentioned by the reviewer, serum coverage is certainly limited but more than 1000 proteins/sample with no depletion of most abundant proteins might be considered a reasonable rank according to the complexity of the sample

The results are interesting and may be of some value to the research community. Although none of the candidates appears to be likely useful as disease status markers, in conjunction with the results of other studies they may be useful for characterising disease process. 

R.- Thank you very much for your comments. We hope indeed that the results provided by our study might contribute to enlarge our knowledge about COVID-19.

Given the lack of data on non-admitted patients >80,  I think that the authors should temper their statements to indicate that there may an age related difference rather than there is.

R.- We thank the reviewer for this comment. We have reduced the emphasis of our statements relative to age related differences in COVID-19 patients

Round 2

Reviewer 1 Report

I would like the authors for their modifications and explanations: the paper is now in a good shape for publication.